# Formononetin Inhibits Mast Cell Degranulation to Ameliorate Compound 48/80-Induced Pseudoallergic Reactions

**DOI:** 10.3390/molecules28135271

**Published:** 2023-07-07

**Authors:** Zi-Wen Zhou, Xue-Yan Zhu, Shu-Ying Li, Si-En Lin, Yu-Han Zhu, Kunmei Ji, Jia-Jie Chen

**Affiliations:** Department of Biochemistry and Molecular Biology, School of Basic Medical Sciences, Shenzhen University Medical School, Shenzhen University, No. 1066 Xueyuan Road, Nanshan District, Shenzhen 518055, China; zhouzw0704@163.com (Z.-W.Z.);

**Keywords:** formononetin, mast cell, non-IgE, compound 48/80, pseudoallergic reaction

## Abstract

Formononetin (FNT) is a plant-derived isoflavone natural product with anti-inflammatory, antioxidant, and anti-allergic properties. We showed previously that FNT inhibits immunoglobulin E (IgE)-dependent mast cell (MC) activation, but the effect of FNT on IgE-independent MC activation is yet unknown. Our aim was to investigate the effects and possible mechanisms of action of FNT on IgE-independent MC activation and pseudoallergic inflammation. We studied the effects of FNT on MC degranulation in vitro with a cell culture model using compound C48/80 to stimulate either mouse bone marrow-derived mast cells (BMMCs) or RBL-2H3 cells. We subsequently measured β-hexosaminase and histamine release, the expression of inflammatory factors, cell morphological changes, and changes in NF-κB signaling. We also studied the effects of FNT in several in vivo murine models of allergic reaction: C48/80-mediated passive cutaneous anaphylaxis (PCA), active systemic anaphylaxis (ASA), and 2,4-dinitrobenzene (DNCB)-induced atopic dermatitis (AD). The results showed that FNT inhibited IgE-independent degranulation of MCs, evaluated by a decrease in the release of β-hexosaminase and histamine and a decreased expression of inflammatory factors. Additionally, FNT reduced cytomorphological elongation and F-actin reorganization and attenuated NF-κB p65 phosphorylation and NF-κB-dependent promoter activity. Moreover, the administration of FNT alleviated pseudoallergic responses in vivo in mouse models of C48/80-stimulated PCA and ASA, and DNCB-induced AD. In conclusion, we suggest that FNT may be a novel anti-allergic drug with great potential to alleviate pseudoallergic responses via the inhibition of IgE-independent MC degranulation and NF-κB signaling.

## 1. Introduction

Allergic diseases are systemic life-threatening disorders that include atopic dermatitis, allergic rhinitis, asthma, chronic urticaria, and food allergies. These inflammatory and allergic reactions are partly regulated via mast cells (MCs), which release intracellular stores of inflammatory mediators (i.e., degranulation) via either canonical immunoglobulin E (IgE)-dependent mechanisms or a hypersensitive IgE-independent reaction [1,2,3,4]. These non-IgE “pseudoallergic” reactions occur when MC degranulation is triggered via ligand binding or the stimulation of a G protein-coupled receptor (GPCR) on the MC membrane; many environmental and ingested substances are ligands or stimuli for Mas-related GPCR X2 (MRGPRX2), a critical receptor on the MC surface [5,6,7]. Two such ligands, compound 48/80 (C48/80) and substance P, directly activate MRGPRX2, thereby initiating a phosphorylation cascade that upregulates inflammatory pathway signaling (e.g., NF-κB). Once activated, MCs release inflammatory mediators (β-hexosaminidase (β-Hex) and histamine) and pro-inflammatory cytokines (tumor necrosis factor (TNF)α and interleukin (IL)-13) that initiate and promote the continuation of allergic inflammation [8]. Thus, the pharmacological inhibition of this MC activation pathway could serve to address IgE-independent allergic diseases, though such strategies have not been thoroughly investigated.

Several lines of evidence indicate that the bioactive isoflavonoid formononetin (FNT, 7-hydroxy-4′-methoxyisoflavone, CAS number 485-72-3) may have considerable anti-inflammatory and anti-allergic effects [9]. FNT is found in a number of common perennial herbs (red clover (*Trifolium pratense*) and Mongolian milkvetch, also known as Huangqi (*Astragalus membranaceus*)) used in traditional Chinese medicine to treat skin and respiratory diseases (e.g., allergic asthma). FNT attenuates atopic dermatitis (AD) by upregulating A20 expression via estrogen GPCR activation and reduces airway inflammation in ovalbumin (OVA)-induced asthmatic mice [10]. Additionally, we previously reported an anti-allergic effect of FNT on IgE-induced MC activation via the disruption of IgE/FcεRI signaling [11], but the effect on non-IgE pseudoallergic MC activation was not determined. After all, since the non-IgE-dependent pathway for MC activation is different from the IgE-dependent pathway, it is valuable to explore the role of FNT in treating C48/80-induced pseudo-anaphylaxis. We report here on the anti-allergic effects of FNT in a mouse model of C48/80-induced pseudoanaphylaxis and in a pathological mouse model of DNCB-induced AD. We also examine the mechanisms of action via in vitro cell culture experiments.

## 2. Result

### 2.1. Inhibition of C48/80-Induced Cell Degranulation in RBL-2H3

The cell viability analysis of FNT (structure in Figure 1A) showed no cytotoxicity against RBL-2H3 at concentrations up to 100 µM (Figure 1B). In a cell culture model of C48/80-induced RBL-2H3 activation, FNT inhibited the release of β-Hex (IC_50_ = 48.24 ± 2.41 μM) and histamine (IC_50_ = 42.38 ± 2.12 μM) in a dose-dependent manner (Figure 1C,D). Additionally, RT-qPCR analysis of these treated cells revealed that the transcriptional levels of C48/80-induced proinflammatory factors (e.g., TNFα and IL-13) were also dose-dependently suppressed via the treatment with FNT (Figure 1E,F).

### 2.2. Inhibition of C48/80-Induced Morphological Changes in RBL-2H3

Toluidine blue or F-actin staining can reveal MC morphology and the release of heterochromatin particles that is characteristic of MC activation [12]. Pretreatment with FNT reduced the appearance of irregularly shaped cells and of cytoplasmic purple particles in RBL-2H3 following C48/80 stimulation (Figure 2A,B). In addition, C48/80-mediated F-actin reassembly in RBL-2H3 was inhibited using FNT treatment (Figure 3A,B). Thus, FNT suppressed morphological changes and cytoskeletal decomposition in a cell culture model of MC activation.

### 2.3. Inhibition of C48/80-Induced NF-κB Activity in RBL-2H3

Western blot analysis revealed that the levels of p-IκBα/IκBα and p-p65/p65 were increased significantly via C48/80 treatment, and FNT blunted this effect in a dose-dependent manner (Figure 3A–C). This result suggests that C48/80-induced MC degranulation proceeds via an MRGPRX2-NF-κB signaling pathway. To confirm the effect on NF-κB signaling, we found that FNT inhibits luciferase reporter activity in an NF-κB-dependent luciferase assay (Figure 3D). These data suggest that FNT attenuates C48/80-induced NF-κB signaling in MCs.

### 2.4. Inhibition of C48/80-Induced Cell Degranulation in Primary BMMCs

The cell viability analysis of FNT showed no cytotoxicity against primary BMMCs at concentrations up to 100 µM (Figure 4A). In a cell culture model of C48/80-induced BMMC activation, FNT inhibited the release of β-Hex (IC_50_ = 50.24 ± 2.51 μM) and histamine (IC_50_ = 59.88 ± 2.99 μM) in a dose-dependent manner (Figure 4B,C). The RT-qPCR analysis of these treated cells revealed that the transcriptional levels of C48/80-induced proinflammatory factors (e.g., TNFα and IL-13) were also dose-dependently suppressed via treatment with FNT (Figure 4D,E).

### 2.5. Attenuation of C48/80-Mediated Pseudoallergic Reactions in the Passive Cutaneous Anaphylaxis (PCA) Mice

As shown in Figure 5A, PCA mice display increased Evans Blue extravasation in the paws, but this effect is blunted in a dose-dependent manner via the pre-treatment with FNT (Figure 5A). The histological analysis of H&E/toluidine blue-stained paw sections revealed that FNT reduced MC recruitment and degranulation to a similar extent as the Dexa-treated positive control (Figure 5B). The FNT pre-treatment also mitigated paw swelling (Figure 5C) and dye extrusion (Figure 5D). Taken together, we conclude that FNT alleviates local cutaneous inflammation in PCA mouse paws.

### 2.6. Attenuation of C48/80-Mediated Pseudoallergic Reactions in the Active Systemic Anaphylaxis (ASA) Mice

We implemented the ASA model, as shown in Figure 6A. We found that C48/80 stimulation led to a significant drop in rectal temperature over 30 min, but FNT treatment blunted this effect (Figure 6B). Furthermore, FNT suppressed the elevated serum histamine levels observed in the ASA model mice (Figure 6C).

### 2.7. Alleviation of DNCB-Induced Atopic Dermatitis Symptoms in Mice

We implemented the DNCB-induced AD model and found that FNT treatment reduced skin lesions and swelling (Figure 7B). The ear thickness of AD mice was markedly reduced in the FNT groups (Figure 7D), and the staining of pathological sections showed decreased infiltration of MCs (Figure 7C,E). Taken together, we conclude that FNT treatment can attenuate local anaphylaxis in a model of ADs. Epidermal skin barrier dysfunction and inflammation play a key role in AD formation [13]. Two proteins from the keratohyalin granules, filaggrin, and loricrin, function in the formation of the epidermal skin barrier [14]. In this present study, FNT prevented filaggrin and loricrine expression reductions in the TNF-α/IFN-γ induced HaCaT cells (Figure 8A–C). It is consistent with the result that FNT alleviates the DNCB-induced AD mouse model.

## 3. Discussion

Growing evidence suggests that the bioactive isoflavone FNT has shown potential for the prevention and treatment of several diseases, including cancer, obesity, and neurodegenerative disease [15,16]. In particular, FNT has shown a unique advantage in anti-inflammatory effects on inhibiting IgE-mediated allergic diseases. For example, La Yi et al. found that FNT alleviated airway inflammation and lung oxidative damage in OVA-induced asthmatic mice [17]. Our previous study also demonstrated that FNT inhibited IgE-dependent MC degranulation [11]. Interestingly, in this current study, we found that FNT suppressed non-IgE-dependent MC activation and its inflammatory reaction, especially C48/80 stimulation. Thus, these findings expand the potential of the application of FNT in the anti-allergic effect.

The overall immune regulating influence of FNT is broad, and the mechanisms of action are not well understood. Previous studies have shown that FNT suppresses inflammatory responses in IgE-induced RBL-2H3 and primary BMMCs [11,18], FNT inhibits pro-inflammatory factor TSLP production in human keratinocyte HaCaT cells [19], ameliorates IL-13-induced inflammation in human nasal epithelial cells JME/CF15 [20], (iv) downregulates NF-kB and JNK signaling, and upregulates the anti-oxidative HO-1 pathway in OVA-induced airway inflammation in asthmatic mice [17], (v) regulates XBP-1 transcription to decrease IgE production in a human B cell line [21], (vi) inhibits USP and diminishes the expression of IgE receptors in MCs [11,22], and (vii) activates the estrogen GPCR, leading to the upregulation of A20 expression in a FITC-induced model of AD [10]. We found here that FNT also inhibits inflammatory responses in the context of non-IgE C48/80-induced MC activation in vitro and suppresses inflammatory responses in vivo in pseudoallergic disease models of PCA, ASA, and DNCB-induced AD. We suggest that FNT may ameliorate these pseudoallergic responses by blocking MC degranulation via the inhibition of the MRGPRX2/NF-kB signaling axis, but this requires further experimental confirmation.

Our study adds to the growing evidence that the bioactive isoflavone FNT has great therapeutic potential in several disease indications [9,23,24,25,26,27], which we expand here to include treatment for pseudoallergic responses. Regarding allergic disease, one study showed that plant secondary metabolites like FNT are enriched in schools with a low risk of asthma symptoms, suggesting that these compounds protect against the allergies associated with exposure to indoor chemicals [28]. Our findings provide an initial mechanistic insight for this observation by connecting FNT with the suppression of IgE-independent mast cell degranulation and pseudoallergic reactions. Thus, FNT can serve as a predictive risk index for the indoor development of allergic disease and also serves as a promising therapeutic lead for treatment.

## 4. Materials and Methods

### 4.1. Materials

FNT (formononetin) was purchased from TargetMol (Boston, MA, USA). 4-nitrophenyl N-acetyl-β-D-glucosaminide was purchased from Sigma (St. Louis, MO, USA). Dexamethasone (Dexa), ketotifen fumarate (Keto), Evans blue, and toluidine blue were purchased from Meilun Biotechnology (Dalian, China). Primary rabbit antibodies against IκBα (ab76429, monoclonal), p-IκBα (ab133462, monoclonal), p65 (ab32536, polyclonal), GAPDH (ab181602, monoclonal), and loricrine (ab176322, monoclonal) were purchased from Abcam (Cambridge, MA, USA). Antibodies targeting filaggrin (DF13853, monoclonal) were purchased from Affinity Biosciences (Cincinnati, OH, USA). Anti-p-p65 (sc-135769, monoclonal), anti-rabbit IgG-HRP (sc-2357, monoclonal), and 4% paraformaldehyde solution in PBS were from Santa Cruz Biotechnology (Dallas, TX, USA). 2,4-dinitrochlorobenzene (DNCB) was purchased from Tokyo Chemical Industry (Tokyo, Japan). C48/80 was purchased from Cayman Chemical (Ann Arbor, MI, USA).

### 4.2. Cell Culture

Rat basophilic leukemia-2H3 cells (RBL-2H3, RBLs, Cellcook Biotechnology, Guangzhou, China) were cultured at 37 °C in Dulbecco’s Modified Eagle’s Medium (DMEM) supplemented with 1% penicillin and streptomycin (P&S) and 10% fetal bovine serum (FBS, Gibco, Grand Island, NY, USA). Mouse bone marrow-derived mast cells (BMMCs) were isolated from mouse bone marrow cells and cultured in RPMI-1640 media supplemented with 10% FBS, 1% P&S, stem cell factor (SCF), and IL-3 [29].

### 4.3. Cell Viability Assay

Cell viability was determined with the CCK-8 Cell Counting Kit 8 (MedChem Express, Monmouth Junction, NJ, USA). RBL-2H3 or BMMCs were treated with a dilution series of FNT and incubated for 24 h, after which CCK8 was added. The absorbance at 450 nm was measured after 2 h using a multi-well plate reader (Bio-Rad, Hercules, CA, USA).

### 4.4. Degranulation Assay

BMMCs or RBL-2H3 cells were treated with a dilution series of FNT or Dexa for 30 min and then were stimulated with C48/80 (15 µg/mL) for 30 min. Degranulation was assayed by measuring β-hex and histamine release as described previously [30]. For β-Hex release assays, substrate solution was mixed with supernatant from each group, and the reaction was stopped by the addition of a Na_2_CO_3_/NaHCO_3_ solution. The absorbance at 405 nm was measured in a plate reader. For histamine release assays, supernatants collected from RBL-2H3 and BMMCs were tested using a commercially available enzyme-linked immunosorbent assay (ELISA; IBL, Hamburg, Germany) according to the manufacturer’s instructions. Half-maximal inhibitory concentrations (IC_50_) were determined for β-hex and histamine release.

### 4.5. Toluidine Blue Staining

RBL-2H3 cells were fixed with paraformaldehyde and stained with toluidine blue solution as described previously [12]. The heterochromatic particles in the stained cells were observed using an inverted microscope (Carl Zeiss, Goettingen, Germany). The C48/80-activated cells and non-activated cells were counted from five different random visual fields.

### 4.6. F-Actin Staining

RBL-2H3 cells were fixed with paraformaldehyde and incubated with FITC-labeled phalloidin solution (Yeasen, Shanghai, China) as described previously [11]. The F-actin-labeled cells were observed under a fluorescence microscope (Carl Zeiss, Goettingen, Germany).

### 4.7. Protein Extraction and Western Blotting

RBL-2H3 cells (6 × 10^5^ cells/well) were grown overnight and treated as described in Cell Culture section. The FNT and Dexa groups were treated with drug for 2 h, after which all groups (except control RBL-2H3) received C48/80 (15 µg/mL) stimulation for 10 min. Protein was extracted with RIPA buffer and quantified with a BCA assay kit (Beyotime, Beijing, China). Anti-p65 (1:10,000), anti-p-p65 (1:1000), anti-IκBα (1:2000), anti-p-IκBα (1:10,000), anti-loricrin (1:1000), anti-filaggrin (1:1000), and anti-GAPDH (1:2000) were used as primary antibodies to incubate the membranes at 4 °C overnight. Secondary antibody (anti-rabbit IgG-HRP, 1:2000) was incubated at room temperature for 1 h. Immunoreactive protein bands were observed with the ChemiDoc Imaging System (Bio-Rad, Hercules, CA, USA) and quantified using Image J software 1.80v (National Institutes of Health, Bethesda, MA, USA).

### 4.8. Luciferase Assay

RBL-2H3 cells (1.2 × 10^5^ cells/well) were incubated until 70–90% confluent and then sequentially transfected using liposome reagent (Bio-Generating, Changzhou, China)) consisting of serum-free medium, plasmids (pGL3-basic, pGL3-control, pNF-κB-Luc and pRL-TK) and liposome. At 48 h post-transfection, we disintegrated the cells using the Luciferase Assay Kit (Promega, Madison, WI, USA) and measured enzymatic activity with a luminometer (Biotek, Biotek, Winooski, VT, USA).

### 4.9. RNA Extraction and RT-qPCR

RBL-2H3 cells were pretreated (with or without pretreatment) with different doses of FNT for 2 h and then stimulated with C48/80 (15 µg/mL) for 2 h before RNA extraction and analysis. Using cells that had been pretreated with drug and C48/80, total RNA was extracted with a RNeasy Mini Kit (Qiagen, Duesseldorf, Germany) according to the manufacturer’s instructions. The concentration and purity of the extracted RNA (A260/A280) were measured, and cDNA was synthesized via reverse transcription of 1 µg total RNA using the HiScript III RT SuperMix (Vazyme, Nanjing, China) in accordance with the manufacturer’s instructions. Gene expression levels were analyzed using real-time RT-qPCR with a TB Green^®^Premix ExTaqTM (Takara, Tokyo, Japan) in a qTOWER 2.2 system (Analytik Jena, Upland, CA, USA). Gene expression levels of target genes were normalized relative to *GAPDH*. The primers were listed in Table 1.

### 4.10. C48/80-Stimulated PCA Model

The C48/80-stimulated PCA model was generated using a modified version of the method described by Ping Zhang et al. [31]. Female BALB/c mice were randomly divided into the following five groups: the control group, model group, FNT group (25, 50 mg/kg), and Dexa group (20 mg/kg). Left paw thickness was measured with vernier calipers (Deli, Zhejiang, China). The FNT and Dexa groups were administered oral doses of drugs normalized for body weight, respectively. The other groups were given the corresponding volume of normal saline. After 1 h, mice in the model group, FNT group, and Dexa group were injected intradermally with 25 µL C48/80 (0.4 mg/mL). The control group was given a corresponding volume of normal saline. After 30 min, 200 µL 1% Evans Blue dye in saline was injected into the tail vein of all groups. The mice were euthanized after 30 min, and the left and right paws were removed. The left paw was photographed for records. The thickness of the left paw after stimulation was measured with vernier calipers, and the degree of paw swelling was calculated. Then, the left paw was soaked in 700 µL formamide and extracted at 65 °C for 12 h. The supernatant was taken, and the dye absorbance was quantitated at 620 nm on a microplate reader (Thermo Fisher Scientific Inc., Waltham, MA, USA). The right paw was fixed by soaking in a 4% formaldehyde solution, embedded in paraffin, and then sliced.
Paw swelling (%) = (paw thickness after stimulation − paw thickness before stimulation)/paw thickness before stimulation × 100%

### 4.11. C48/80-Induced ASA Model

The C48/80-induced ASA model was generated using a modified version of the method described by Ping Zhang et al. [31]. Female BALB/c mice were randomly divided into the following five groups: the control, model, FNT groups (25 and 50 mg/kg) and Dexa group (20 mg/kg), and the FNT groups. Dexa, a glucocorticoid receptor agonist, was reported to attenuate the C48/80-induced allergic reactions in vivo [32]. After 30 min, C48/80 (0.25 mg/kg) was injected through the tail vein in the model group, while FNT, Dexa, and control groups were injected with the same volume of normal saline. The rectal temperatures of the mice were measured immediately and for 30 min after C48/80 injection. Blood was obtained from the orbital venous plexus and submitted to serum histamine quantitation via ELISA (kits from IBL, Hamburg, Germany).

### 4.12. DNCB-Induced AD Model

The AD model was generated using a modified version of the method described in the following literature [33]. Female BALB/c mice were randomly divided into the following five groups (n = 6): the control group, AD group, FNT (25 mg/kg), FNT (50 mg/kg), and Keto (50 mg/kg). Keto is an H1 receptor antagonist and MC stabilizer and is used as a positive control treatment for inhibition of activated MCs in vivo [34]. House dust mite (HDM) plays an exacerbator role in AD skin model [35]. In all groups except the control group, 100 µL mixture, including 1% DNCB (dissolved in PEG) and HDM (1 mg/mL, dissolved in PBS), was evenly applied to sensitize the right ear twice a week. After two rounds of stimulation, mice in the administration group were intraperitoneally injected with corresponding doses of FNT and Keto twice a week, respectively. Ear thickness was measured with vernier calipers every four days after DNCB/HDM induction for 24 h. See Figure 7A for details. After 35 days of treatment, the right ear was photographed for recording and then removed. The right ear was soaked in 4% formaldehyde solution and fixed and then embedded with paraffin for tissue slicing. The number of MCs was counted from five different random sites in each group.

### 4.13. Murine Models and Histology

Paw specimens from PCA mice or ear specimens from AD mice were prepared and stained with hematoxylin, eosin, and toluidine blue. We evaluated Evans Blue dye extraversion and paw swelling for PCA or ear thickness for AD.

### 4.14. Statistical Analysis

The data are expressed as the mean ± standard deviation (SD). We used one-way analyses of variance (ANOVAs) to compare the differences between groups. All analyses were conducted in Prism 8 (GraphPad, La Jolla, CA, USA). We consider *p* < 0.05 to be statistically significant.

## 5. Conclusions

This study demonstrates that FNT inhibits C48/80-induced MC activation, primarily by reducing the release of histamine and β-hex and suppressing the expression of inflammatory factors (TNFα and IL-13) via NF-κB pathway inhibition. The oral administration of FNT alleviates the allergic inflammation in C48/80-induced PCA and ASA mice. In addition, FNT also reduced DNCB-induced AD. We suggest that FNT is a promising candidate for addressing pseudoallergic reactions by inhibiting IgE-independent MC activation.

## Figures and Tables

**Figure 1 molecules-28-05271-f001:**
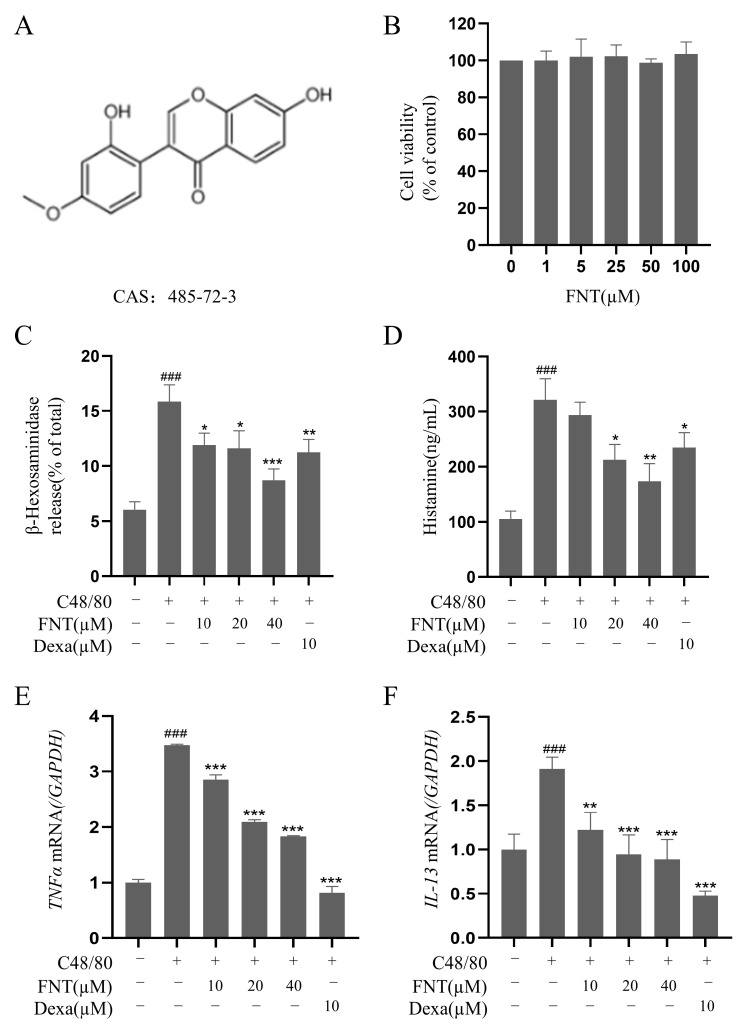
FNT inhibits C48/80-induced cell degranulation in RBL-2H3. (**A**) Chemical structure of FNT. (**B**) Cell viability of RBL-2H3 cells treated with 0–100 µM FNT was determined using CCK-8 assays. (**C**–**F**) The inhibitory effect of FNT on the secretion of β-Hex (**C**), histamine (**D**), and the transcriptional level of proinflammatory factors TNFα (**E**) and IL-13 (**F**) in C48/80-mediated RBL-2H3 cells. The data are expressed as the mean ± SDs (n = 3). ### *p* < 0.001 vs. the control group; * *p* < 0.05, ** *p* < 0.01, and *** *p* < 0.001 vs. the model group. FNT, formononetin; β-Hex, β-hexosaminidase; HIS, histamine; Dexa, dexamethasone.

**Figure 2 molecules-28-05271-f002:**
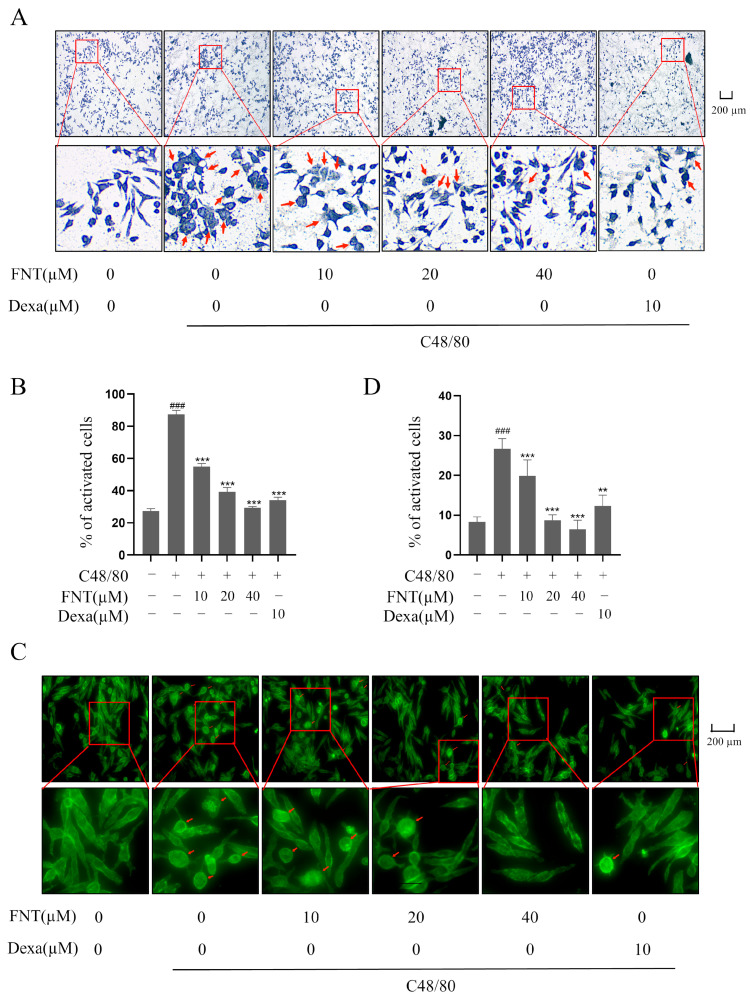
FNT inhibits C48/80-mediated morphological changes in RBL-2H3 cells. RBL-2H3 cells were pre-treated with or without the indicated doses of FNT for 2 h and then stimulated with C48/80 for 10 min. (**A**,**B**) Representative morphological changes in RBL-2H3 cells were shown via toluidine blue staining. Red arrows indicate the cells that become irregular in shape and release purple particles. (**C**,**D**) FITC-phalloidin staining showed morphological changes in cells. F-actin cytoskeleton was stained with green fluorescence. Red arrow indicates oval cells in response to F-actin cytoskeleton changes. Means ± SDs (n = 3) are shown; ### *p* < 0.001 vs. the control group; ** *p* < 0.01, and *** *p* < 0.001 vs. the model group. Abbreviations as in Figure 1.

**Figure 3 molecules-28-05271-f003:**
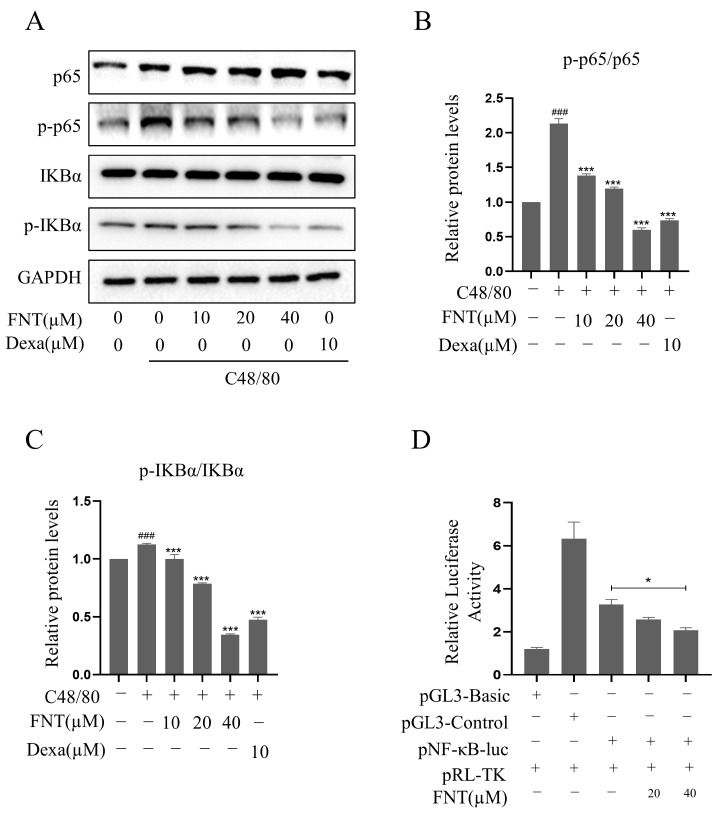
FNT inhibits C48/80-increased NF-κB activity in RBL-2H3 cells. RBL-2H3 cells were pre-treated (or not) with different doses of FNT for 2 h separately and then stimulated with C48/80 for 10 min, the whole cell lysate was taken. (**A**) Western blot analysis of NF-κB pathway signaling molecules (p-p65, p65, p-IκBα, and IκBα) in C48/80-stimulated RBL-2H3 cells treated with 10, 20, or 40 µM FNT. (**B**,**C**) Quantification of the proteins. (**D**) RBL-2H3 cells transfected with pNF-κB-Luc plasmid were treated with FNT for 4h, and the selective inhibition of NF-κB pathway via FNT was detected using dual luciferase reporter gene assay. Means ± SDs (n = 3) are shown; ### *p* < 0.001 vs. the control group; * *p* < 0.05, and *** *p* < 0.001 vs. the model group. Abbreviations as in Figure 1.

**Figure 4 molecules-28-05271-f004:**
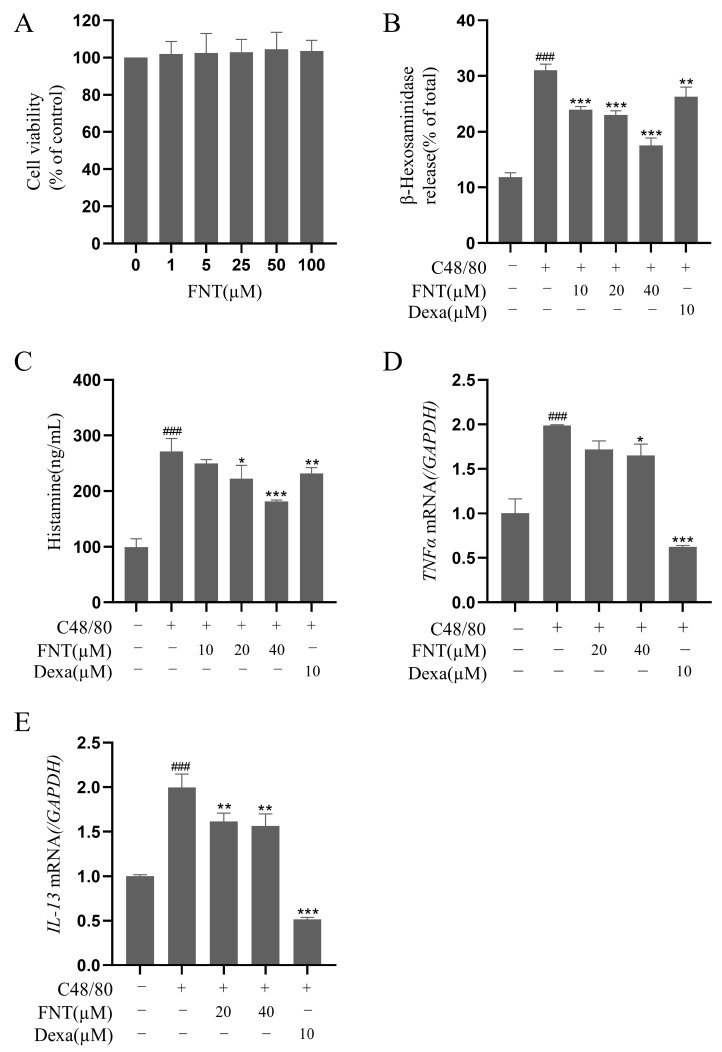
FNT suppresses C48/80-induced cell degranulation in primary BMMCs. (**A**) Cell viability of BMMCs treated with 0–100 µM FNT was determined using CCK-8 assays. (**B**–**E**) The inhibitory effect of FNT on the secretion of β-Hex (**C**), histamine (**D**), and RBL-2H3 cells were pre-treated (or not) with different doses of FNT for 2 h separately and then stimulated with C48/80 for 2h. and the transcriptional level of inflammatory factors TNFα (**E**) and IL-13 (**F**) in C48/80-stimulated BMMCs. The data are expressed as the mean ± SDs (n = 3). ### *p* < 0.001 vs. the control group; * *p* < 0.05, ** *p* < 0.01, and *** *p* < 0.001 vs. the model group. Abbreviations as in Figure 1.

**Figure 5 molecules-28-05271-f005:**
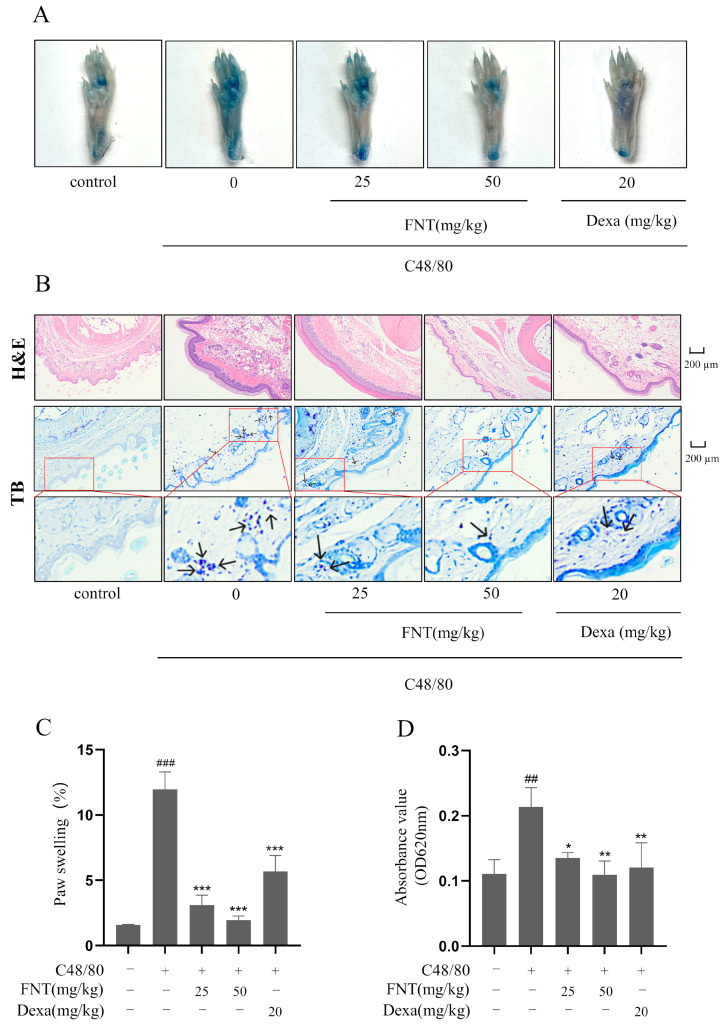
FNT attenuates C48/80-mediated pseudoallergic reactions in PCA Mice. (**A**) Representative images showed the diffusion of Evans blue from paws of PCA mice. (**B**) Representative pictures of pathological changes in PCA mice. H&E staining showed the degree of telangiectasia, and toluidine blue staining showed MC infiltration. Small blue particles indicated by arrows in toluidine blue staining are infiltrating mast cells. (**C**) Evans blue diffusion quantified at 620 nm. (**D**) Quantification of the change in paw thickness after stimulation. The data are shown as the mean ± SDs (n = 6); ## *p* < 0.01, ### *p* < 0.001 vs. the control group, * *p* < 0.05, ** *p* < 0.01, *** *p* < 0.001 vs. the model group. Abbreviations as in Figure 1.

**Figure 6 molecules-28-05271-f006:**
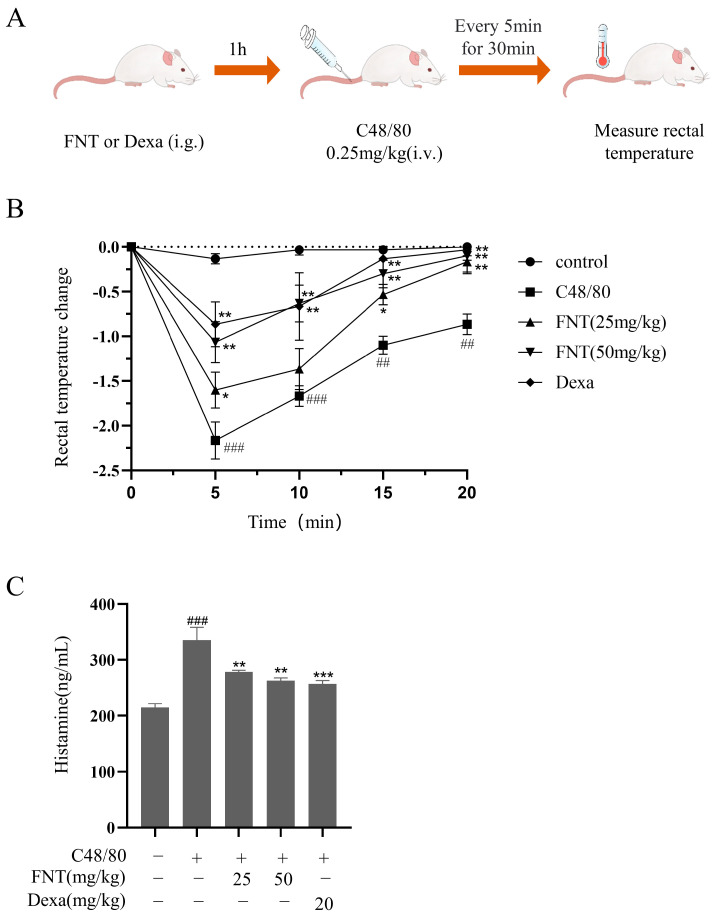
FNT attenuated C48/80-mediated pseudoallergic reactions in ASA mice. (**A**) The protocol for ASA experiment. (**B**) Changes in rectal temperature after injecting with C48/80 were measured every 5 min over the next 30 min in each group. (**C**) Effect of FNT on the release of histamine in ASA mice (determined using ELISA). The data are shown as the mean ± SDs (n = 6); ## *p* < 0.01, ### *p* < 0.001 vs. the control group, * *p* < 0.05, ** *p* < 0.01, *** *p* < 0.001 vs. the model group. Abbreviations as in Figure 1.

**Figure 7 molecules-28-05271-f007:**
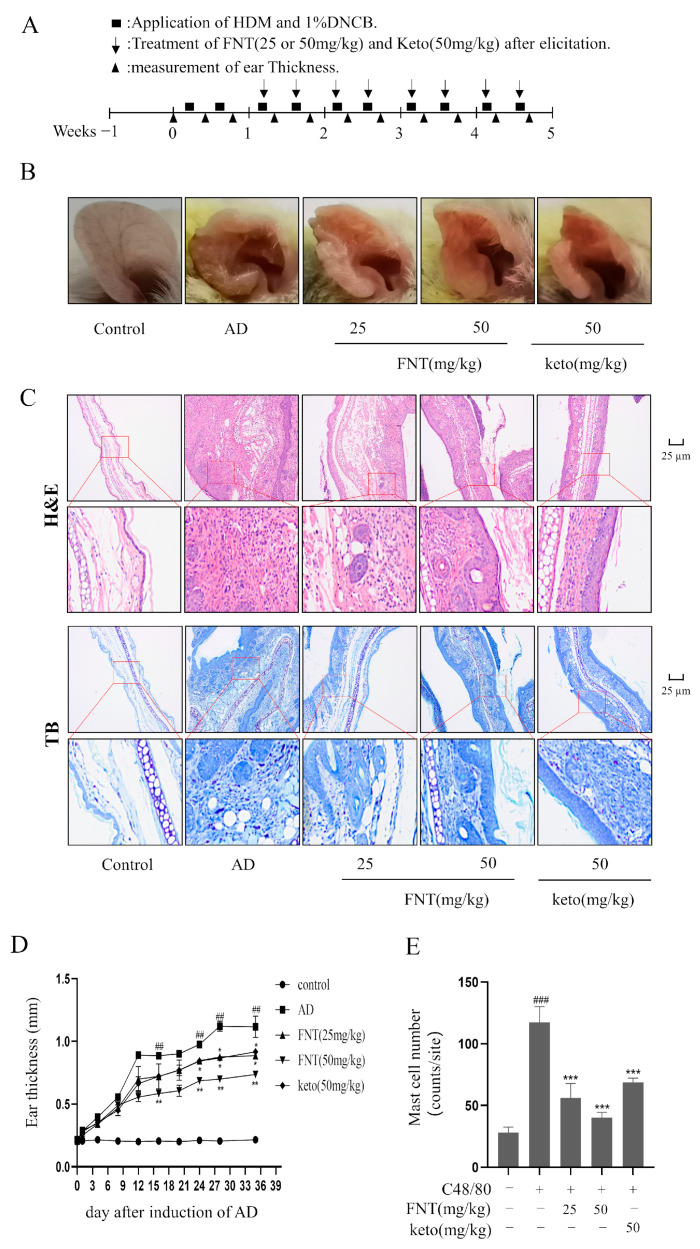
FNT alleviated DNCB-induced local cutaneous inflammation in AD mice. (**A**) Schematic of the protocol for FNT to prevent sensitization and elicitation in experimental HDM and 1% DNCB-induced AD lesions. (**B**) Representative pictures showing the skin lesions from ears. (**C**) Representative pictures of pathological changes in AD mice. H&E staining showed the degree of thrombocytopenia, vascular permeability, and vascular epidermis thickness, and toluidine blue staining showed MCs infiltration. (**D**) The change in ear thickness after induction of AD. (**E**) The MC numbers were counted from five different random sites in each group. The data are shown as the mean ± SDs (n = 6); ## *p* < 0.01, ### *p* < 0.001 vs. the control group, * *p* < 0.05, ** *p* < 0.05, *** *p* < 0.001 vs. the model group. Abbreviations as in Figure 1.

**Figure 8 molecules-28-05271-f008:**
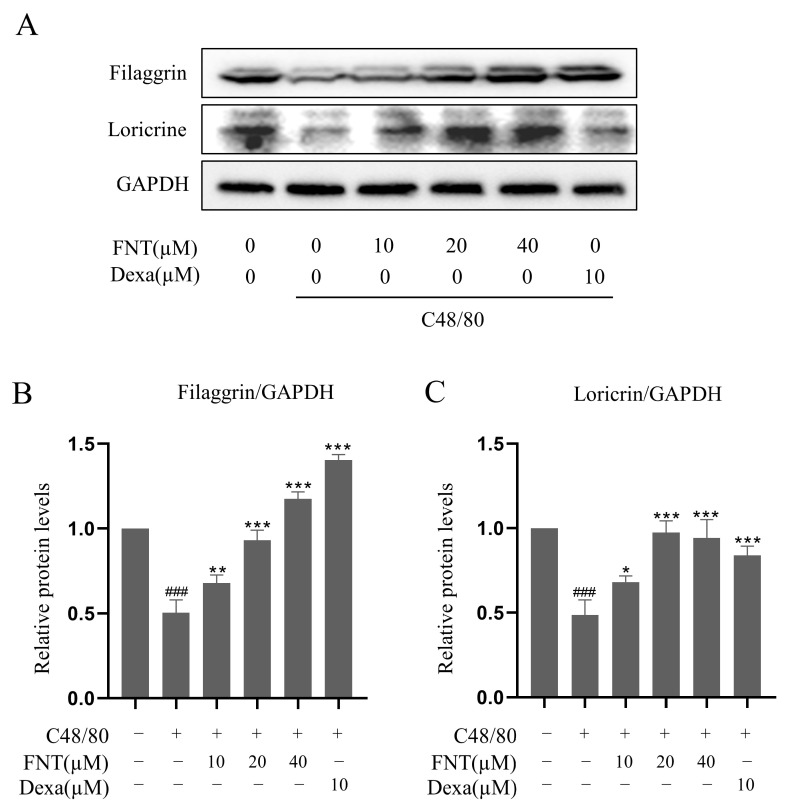
Effect of FNT on the expression of filaggrin and loricrine protein in TNF-α/IFN-γ-stimulated HaCaT cells. HaCaT cells were pretreated with the indicated FNT (10, 20, and 40 µM) or Dexa (20 µM) for 1 h, followed by TNF-α (10 ng/mL) and IFN-γ (10 ng/mL) treatment for 16 h. Whole-cell lysates were then performed for Western blot analysis. (**A**) FNT increased the expression of filaggrin and loricrine. (**B**,**C**) Quantified values of filaggrin and loricrine. The levels of filaggrin and filaggrin were normalized to that of GAPDH. The results are expressed as the mean ± SD (n = 3). ### *p* < 0.001 vs. normal; * *p* < 0.05, ** *p* < 0.01 and *** *p* < 0.001 vs. control. Dexa, the positive control, was used at 10 µM.

**Table 1 molecules-28-05271-t001:** Sequences of primers used in RT-qPCR.

Gene	Forward (5′ to 3′)	Reverse (5′ to 3′)
*Rat GAPDH*	GGCACAGTCAAGGCTGAGAATG	ATGGTGGTGAAGACGCCAGTA
*Rat IL-13*	AGCAACATCACACAAGACC	GGTTACAGAGGCCATTCA
*Rat TNF*α	CCCTGTTCTGCTTTCTCA	GTTCTCCGTGGTGTTCCT
*Mouse GAPDH*	AAGAAGGTGGTGAAGCAGG	GAAGGTGGAAGAGTGGGAGT
*Mouse* *TNFA*	CGTGGAACTGGCAGAAGAG	GTAGACAGAAGAGCGTGGTG
*Mouse IL-13*	CTCTTGCTTGCCTTGGTGGTC	AGGGGAGTCTGGTCTTGTGTGAT

## Data Availability

The datasets used and/or analyzed during this current study are available from the corresponding author upon reasonable request.

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
