# Peer review of "Formononetin Inhibits Mast Cell Degranulation to Ameliorate Compound 48/80-Induced Pseudoallergic Reactions"

_molecules, 2023, doi:10.3390/molecules28135271_

Round 1

Reviewer 1 Report (Previous Reviewer 1)

The authors thoroughly revised the manuscript and answered all questions. I have no questions anymore

Author Response

Thanks for your constructive comments. 

Reviewer 2 Report (Previous Reviewer 2)

Although the authors commented to my comments, the authors still did not give enough information of 48/80.  The concentration of 48/80 used  in in vitro experiments should be described in Method "Degranulation assay".

Author Response

Reply:Thanks for your comments. C48/80 Specific information is presented in the section of Methods. We supplemented the specific concentration of C48/80 in Method "Degranulation assay". Details are as follows:

In the revised section of Methods:

“C48/80 was purchased from Cayman Chemical (Ann Arbor, Michigan, USA).”  “BMMCs or RBL-2H3 cells were treated with a dilution series of FNT or Dexa for 30 min, and then were stimulated with C48/80(15µg/mL) for 30 min.”

This manuscript is a resubmission of an earlier submission. The following is a list of the peer review reports and author responses from that submission.

Round 1

Reviewer 1 Report

In their manuscript Zhou et al show, that the plant derived isoflavone formononetin inhibits pseudoallergic reactions in vivo by reducing mast cell degranulation and inflammation. Although the study is interesting for possible interference with allergic/pseudoallergic reactions the study has considerable deficiencies and must be thoroughly revised.

Specific points:

The terms PSA, ASA and AD should be written out at least somewhere in the manuscript

Also, the cell line RBL-2H3 should be written like this

In the materials and method section and in the supplementary files there is the sensitization phase of both, the ASA and the AD model missing; If done in this way, both models would not work. AD model: is it really 5%DNCB for challenge. It should be the challenge reaction according to the reference; 5%DNCB is a really high concentration, also in combination with the HDM which should induce a severe inflammation or even wounding of the ear. What is “keto”, this is not explained, also HDM needs to be explained.

Also the analysis part of the murine models is only poorly described

Are the ASA and the AD model really pseudoallergic?

The reduction of allergy symptoms by formononetin in model of ASA is also published in Food Funct2023 Mar 20;14(6):2857-2869.doi: 10.1039/d2fo03997d by the authors, what is the difference for the model used in this manuscript?

Figure 1:

Which time points were chosen for stimulation of the cells for the RNA analysis? The induction of either TNF and IL-13 is only very low. In addition, IL-13 is not really a proinflammatory cytokine, it is a Th2 cytokine. Time points are also missing for figure 4D and E

Figure 2 and description of Figure 2 in the results section:

What are irregularly shaped cells, and how is the number of activated cells calculated? Are the irregularly shaped the activated ones? Is here mast cell degranulation visible in the cytochemistry and staining with toluidine blue? Are there degranulated cells visible?

Although there were higher magnifications of the cytochemistry shown, degranulation or the shape of the cells is only visible if the file is further enlarged.

Figure 3 and description of Figure 3 in the results section:

Figure 3A: it should be noted that this is a representative Western Blot. Why are the cells for Western analysis stimulated for 2h with the compounds and 30 min for degranulation and histamine release?

How long were the stimulation phases of the luciferase assay shown in Figure 3D?

Figure 5 and description of Figure 5 in the results section:

Figure 5B: What is shown in the H&E stainings? It seems that there are changes in epidermal thickness after C48/80, is that true? If yes, this is not written in the text. In addition, I don’t see telangiectasia in the histology, why should this be visible in an H&E stain?

Figure 7 and description of Figure 7 in the results section:

The histologic pictures show completely different parts of the ear. In addition, here it is clearly seen that it is dermal swelling or increased dermal thickness. This has to be explained in more detail. The authors say that there are more eosinophils. How were they stained? Can this be quantified?

english language is ok

Reviewer 2 Report

In this paper, the authors examined the effects of FNT on 48/80-induced activation of mast cells and allergic responses in mice.  They found that FNT inhibited potently 48/80-induced pseudoallergic responses.  

Major comments:

1) The authors should clarify the effects of FNT was selective to 48/80-induced reactions.  For example, the authors should compare the IC50 with the antigen-induced degranulation.  The authors should discuss the mechanisms buy which FNT inhibited 48/80-induced responses.

2) In generally, RBL-2H3 cells and BMMC were not responsed to 48/80 because they are mucosal type mast cells.  What the concentration of 48/80 used? The information was important.

3) Usually, the stimulation induced the reduction of IkB proteins. However, in fig. 3, there was no reduction of IkB.  The reason should be explained.

4) FNT has phenolic OH.  Did this compound bind to 48/80 directly and inhibit the activity?

5) Line 151, the authors referred the infiltration of eosinophils.  The authors should count it.

Minor points:

1) Fig. 3A shows 48/80 treatments is only 4 groups. Is it true? 

2) The authors should add the comments the effects of dexa and 

keto.  
